# Comparative Kinetics of Acetyl- and Butyryl-Cholinesterase Inhibition by Green Tea Catechins|Relevance to the Symptomatic Treatment of Alzheimer’s Disease

**DOI:** 10.3390/nu12041090

**Published:** 2020-04-15

**Authors:** Edward J. Okello, Joshua Mather

**Affiliations:** 1Human Nutrition Research Centre, Population Health Sciences Institute, Faculty of Medical Sciences, Leech Building, Framlington Place, Newcastle upon Tyne NE2 4HH, UK; 2School of Biomedical, Nutritional and Sport Sciences, Faculty of Medical Sciences, Catherine Cookson Building, Newcastle University, Framlington Place, Newcastle upon Tyne NE2 4HH, UK; J.Mather3@ncl.ac.uk

**Keywords:** Alzheimer’s, cholinesterase inhibitors, catechins, flavan-3-ols, acetylcholine, neurodegeneration

## Abstract

Alzheimer’s disease (AD) is characterised by the apoptosis of cholinergic neurons and the consequent attenuation of acetylcholine mediated neurotransmission, resulting in neurodegeneration. Acetyl-cholinesterase (AChE) and butyryl-cholinesterase (BuChE) are attractive therapeutic targets in the treatment of AD since inhibition of these enzymes can be used to restore synaptic concentrations of acetylcholine. Whilst inhibitors for these enzymes such as galantamine and rivastigmine have been approved for use, none are able to halt the progression of AD and are responsible for the production of troublesome side-effects. Efficacious cholinesterase inhibitors have been isolated from natural plant-based compounds with many demonstrating additional benefits beyond cholinesterase inhibition, such as antioxidation and anti-inflammation, which are key parts of AD pathology. In this study, five natural flavan-3-ol (catechin) compounds: ((-)-epicatechin (EC), catechin, (-)-epicatechin-3-gallate (ECG),) (-)-epigallocatechin (EGC), (-)-epigallocatechin-3-gallate (EGCG), isolated from green tea, were screened for their cholinesterase inhibitory activity using the Ellman assay. The kinetics of inhibition was determined using reciprocal Lineweaver-Burk plots. EGCG was the only compound found to produce statistically significant, competitive inhibition, of both AChE (*p* < 0.01) and BuChE (*p* < 0.01) with IC50 values of 0.0148 µmol/mL and 0.0251 µmol/mL respectively. These results, combined with previously identified antioxidative and anti-inflammatory properties, highlight the potential use of EGCG in the treatment of AD, provided it can be delivered to cholinergic neurons in therapeutic concentrations. Further testing of EGCG in vivo is recommended to fully characterise the pharmacokinetic properties, optimal method of administration and efficacy of this novel plant-based compound.

## 1. Introduction

Alzheimer’s disease is the most common form of dementia in the UK accounting for 50–75% of cases [1]. Indeed, the number of people with AD in the UK is expected to nearly double by 2040 to 1.6 million patients, with the total cost of treating dementia expected to reach £94.1 bn by 2040 [2]. AD is an irreversible neurodegenerative disorder in which there is a progressive and continual deterioration in cognitive function [3]. Initial symptoms typically include confusion, repetitive questioning and changes in mood [4]. These symptoms progressively worsen with patients beginning to experience delusion and aphasia until eventually suffering from difficulties with breathing, eating and moving, especially without significant assistance from carers [4].

The aetiology and pathophysiology of AD is complex and is characterized by the apoptosis of cholinergic neurons, especially those in the limbic and neocortical regions [5] and the consequent attenuation of acetylcholine mediated neurotransmission. This cholinergic neuronal apoptosis is thought to result primarily from the abnormal production and processing of the β-amyloid and tau proteins [6,7,8,9,10]. In addition to this, there is increasing recognition that the immune system, and the subsequent inflammatory cascades it stimulates, are also of significant importance [11]. Genetic susceptibility in the form of polymorphism at the APOE gene locus also appears to modulate risk with the ε4 variant conferring greater risk of AD development [12].

Consequently, there is considerable interest in the development of effective treatments to alleviate the symptoms of AD and potentially halt the underlying neurodegeneration the disease causes. Such treatment should lead to an improved quality of life for patients whilst also reducing the financial burden that health systems experience as a result of the disease. The loss of cholinergic neurons, especially those in the limbic and neocortical regions (which are vital for cognitive functions such as memory, learning and attention [13] leads to diminished acetylcholine (ACh) production [5]. Therefore, inhibition of Acetylcholinesterase (AChE), the enzyme responsible for ACh metabolism [14], is a key target in the treatment of AD. Effective inhibition of AChE allows synaptic levels of ACh to be restored, thus alleviating the symptoms of AD.

In addition, Butyrylcholinesterase (BuChE) has also been identified as another important cholinesterase relevant to AD [15]. Levels of BuChE expression are increased by up to 120% in patients with AD, possibly due to compensatory mechanisms which relate to AChE expression which can be reduced by 55–67% [15]. Consequently, the majority of ACh is metabolised by BuChE in the later stages of AD.

Therefore, dual inhibition of BuChE, as well as AChE, would be an attractive property for any potential novel cholinesterase inhibitor. Several AChE inhibitors including galantamine (Gal), donepezil and rivastigmine have been developed and approved for the treatment of mild AD [16]. Only rivastigmine has been approved for the inhibition of both AChE and BuChE [16], although Gal has been shown to inhibit BuChE to a certain extent [17]. Gal is currently the treatment of choice for AD and has a dual mechanism of action, increasing the sensitivity of postsynaptic NMDA receptors as well as inhibiting AChE [18]. Both of these factors work to enhance cholinergic neurotransmission. However, despite its efficacy, Gal results in unpleasant side-effects which include cardiac arrhythmias, GI irritation and tremor [19]. These side-effects combined with the fact that Gal is unable to halt the underlying progression of AD means that there is significant interest in the identification of superior alternatives concerning both efficacy and side-effect profile.

Multiple compounds with anti-cholinesterase activity have been derived from natural plant-based sources [20]. These natural compounds often have multiple, wide-ranging health benefits beyond cholinesterase inhibition such as antioxidation and anti-inflammation.

Consequently, natural compounds offer an attractive proposition for the identification of effective AD treatments since they have the potential to improve symptoms by targeting multiple parts of AD pathology, including inflammation and the generation of reactive oxygen species [11].

Flavan-3-ols are an example of a class of naturally occurring compounds, found mostly in green tea, which have been shown to possess antioxidant and anti-inflammatory properties [21]. The average cup of green tea is estimated to contain a total of 67 mg of these flavan-3-ol compounds [22].

In addition, studies in human clinical trials have demonstrated the ability of flavan-3-ol compounds to slow cognitive decline [23]. Consequently, flavan-3-ols are an attractive source from which to identify novel cholinesterase inhibitors. Table 1 summarises the five flavan-3-ol compounds which were investigated in this study along with Gal, the gold standard for AChE inhibition.

Based on this information, it was hypothesised that the beneficial effects of these flavan-3-ol compounds extended to cholinesterase inhibition.

## 2. Materials and Methods

Acetylcholinesterase (AChE) (EC 3.1.1.7) from electric eel, Butyrylcholinesterase (BuChE) (EC 3.1.18), from equine serum, acetylthiocholine iodide (ATC), butyrylthiocholine iodide (BTC), 5:5-dithiobis-2-nitrobenzoic acid (DTNB), sodium phosphate (mono and dibasic), sodium bicarbonate, Galanthamine hydrobromide from Lycoris sp, (-)-epicatechin (EC), catechin, (-)-epicatechin-3-gallate (ECG), (-)-epigallocatechin (EGC) and (-)-epigallocatechin-3-gallate (EGCG) were purchased from Sigma-Aldrich (UK).

### 2.1. Ellman Essay

The Ellman assay was used to quantify inhibition of the cholinesterase enzymes [26] with some modification [27]. In brief, 5 µL of AChE isolated from the electric eel (0.03 U/mL, pH 8 Na_3_PO_4_ buffer) or 5 µL of BuChE isolated from equine serum (0.03 U/mL, pH 8 Na_3_PO_4_ buffer) was added to a generic 96 well plate followed by 200 µL of pH 8 Na_3_PO_4_ buffer, 5 µL of DTNB (0.3 mM in pH 7 Na_3_PO_4_ buffer) and finally, 5 µL of the inhibitor under investigation (dissolved in de-ionised H_2_O). Blank (no inhibitor or enzyme) and control (no inhibitor) samples were also set up with the total volume made up to 215 µL using pH 8 phosphate buffer prior to incubation. Samples were then mixed for 10 s at 600 rpm and incubated at 30 °C for 10 min in the Ascent Multiskan Plate Reader. Following this, 5 µL of the substrate ATC (0.5 mM in pH 8 Na_3_PO_4_ buffer) or BTC was added (0.5 mM in pH 8 Na_3_PO_4_ buffer) to each well.

Average absorbance was then measured every 30 s for 6 min at 405 nm. The concentration of each inhibitor used to initially test for cholinesterase inhibition varied according to solubility (Table 2).

A test for synergy between four of the flavan-3-ols under investigation was also carried out. Epicatechin (EC), catechin, epigallocatechin (EGC) and epigallocatechin gallate (EGCG) were mixed in a 1:1:1:1 ratio resulting in a test concentration for each inhibitor of 1.25 mg/mL in the synergy mixture.

Average absorbance from the blank wells was calculated and subtracted from each control and inhibitor concentration triplicate absorbance average. Percentage inhibition was then calculated using the following equation: (mean absorbance with inhibitor/negative control mean absorbance) × 100.

If statistically significant inhibition was observed, a serial dilution was carried out to produce a range of concentrations from which a concentration-percentage inhibition curve could be constructed using SigmaPlot^TM^ (Systat Software Inc., San Jose, CA, USA) From these plots, the IC_50_ value was calculated.

### 2.2. Kinetic Analysis: Lineweaver–Burk Plots (L-B)

Kinetics of inhibition was determined using L-B reciprocal plots which were constructed using SigmaPlot^TM^. Two concentrations of each inhibitor, as well as a control, were tested for their anti-cholinesterase activity using four different substrate concentrations (0.5, 0.25, 0.125 and 0.0625 mM).

Results were then plotted on a graph of 1/absorbance against 1/[substrate] and lines were extrapolated backwards to determine the point at which the three lines intersected. The point of intersection allows the type of inhibition present to be identified. From these plots, the K_M_ and V_MAX_ for each inhibitor concentration were calculated by obtaining the gradient and y-intercept for each trend line from Excel. V_MAX_ was calculated by 1/Y-intercept and K_M_ was calculated by multiplying the gradient and the V_MAX_.

Therefore, for non-competitive inhibitors, the V_MAX_ will be reduced, and the K_M_ will remain the same—the opposite is true for competitive inhibition [28]. For uncompetitive inhibition, there is a reduction in both the V_MAX_ and the K_M_ values whilst for mixed inhibition, V_MAX_ decreases and the K_M_ can either increase or decrease [28].

## 3. Results

### 3.1. AChE Inhibition

Percentage inhibition of AChE in the presence of the compounds under investigation is shown in Table 3.

Of the seven inhibitors tested, five exhibited statistically significant inhibition (*p* < 0.01), with Gal showing the most potent AChE inhibitory activity (Table 3). In addition to Gal, ECG, EGC and EGCG were found to possess significant AChE inhibitory properties and were studied further.

Synergism between four of the flavan-3-ol compounds (EC, catechin, EGC and EGCG) was observed despite two of the flavan-3-ols, EC and catechin, being identified as inactive when administered alone. EC and catechin and were found to have no significant inhibition of AChE even at the highest available concentrations. Therefore, these compounds were not investigated any further.

Serial dilutions for each of the active compounds and the synergistic mixture yielded a range of concentrations from which IC_50_ values were calculated (Table 4). Whilst ECG did show statistically significant inhibition, its potency was too low for an IC_50_ value to be accurately calculated.

Gal was by far the most potent inhibitor tested with an IC_50_ value 73.2× and 31.8× more potent than EGC and EGCG respectively. The IC_50_ value for the synergistic mixture is expressed in mg/mL and compared with Gal in Table 5.

Again, Gal demonstrates much greater potency with an IC_50_ value 40.8-fold smaller than that of the synergy mixture.

A one-way ANOVA with post-hoc Tukey analysis was then used to reveal the minimum inhibitor concentration required for each compound to achieve statistically significant inhibition of AChE (Table 6).

The inhibitor concentration threshold required for AChE inhibition varied between compounds with Gal producing the lowest concentration required for AChE inhibition. Below these concentrations, no statistically significant inhibition of AChE was detected. The following dose-response curves for each of the active compounds along with the corresponding IC_50_ values are shown in Figure 1.

From this data, it was concluded that of the compounds tested, EGCG was the most efficacious novel cholinesterase inhibitor with the most potent IC_50_ value.

Some synergistic effects were observed with similar percentage inhibition obtained for the synergistic mixture (99.22%) and EGCG (82.35%) despite concentrations of the inhibitors being four times lower in the synergy mixture than when the compounds were tested individually. Despite these findings, none of the novel compounds or the synergy mixture was found to inhibit AChE to the same extent as Gal.

### 3.2. AChE Inhibition Kinetics

The kinetics of AChE inhibition was then determined for each of the active compounds using the reciprocal L-B plots shown in Figure 2.

From these plots, the V_MAX_ and K_M_ were calculated (Table 7), allowing the type of enzyme inhibition present to be confirmed.

Gal and EGCG were found to competitively inhibit AChE (Figure 1, Table 7), whereas ECG and EGC were found to be uncompetitive inhibitors of AChE.

### 3.3. BuChE Inhibition

Compounds were then tested for their ability to inhibit BuChE. Percentage inhibition and the corresponding *p*-values obtained from two-sample t-tests are shown in Table 8.

Gal and EGCG were the only compounds that showed extensive inhibition of BuChE, which was high enough for IC_50_ values to be calculated. Whilst ECG (45.65%), EGC (47.72%) and the synergy mixture (56.02%) did show statistically significant inhibition, this extent of inhibition at high concentrations is unlikely to have any relevance clinically. Consequently, these inhibitors were not investigated any further concerning BuChE inhibition.

IC_50_ values were then calculated and again Gal was found to be more potent than EGCG in terms of BuChE inhibition (Table 9).

Gal was 2.51× more potent than EGCG. This is a much smaller difference, suggesting reduced Gal affinity for the BuChE enzyme. 

A one-way ANOVA with post-hoc Tukey analysis revealed the minimum inhibitor concentration required for statistically significant inhibition BuChE to be achieved (Table 10).

The lowest concentration of EGCG tested was sufficient to produce significant inhibition. Consequently, a higher minimum concentration of Gal was required to achieve statistically significant inhibition of BuChE. The IC_50_ plots for BuChE inhibition with Gal and EGCG are shown in Figure 3.

EGCG was identified as the most effective novel inhibitor of BuChE. Although EGCG was less potent than Gal, the difference in the amount of BuChE inhibition produced is relatively small.

### 3.4. BuChE Inhibition Kinetics

The kinetics of BuChE inhibition was again determined using L-B plots (Figure 4).

Gal appears to exhibit mixed inhibition (Figure 4A, Table 11). In contrast, EGCG demonstrates competitive inhibition (Figure 4B, Table 11).

### 3.5. Cholinesterase Affinity Comparison

EGCG was the only novel compound identified as a significant inhibitor of both cholinesterase enzymes. Varying affinity of EGCG for the AChE and BuChE enzymes was identified (Table 12).

Calculated IC_50_ values demonstrated that Gal was 21.5-fold more selective for AChE than BuChE. A similar but much smaller difference in selectivity was identified for EGCG which had an AChE IC_50_ value 1.7-fold smaller than the corresponding BuChE IC_50_ value.

Finally, synergism was observed between flavan-3-ol compounds, producing statistically significant inhibition of both AChE and BuChE (Table 13).

Synergism was more pronounced for inhibition of AChE, resulting in 43.20% more inhibition than for BuChE.

## 4. Discussion

### 4.1. Findings

Analysis of six novel plant-based compounds using the Ellman assay allowed for the identification of significant AChE inhibitory properties in two of the novel compounds studied, EGC and EGCG, and significant BuChE inhibitory properties in one compound, EGCG.

Of the flavan-3-ols investigated, only EGCG possessed significant inhibition of both cholinesterase enzymes. Inhibition of AChE and BuChE is likely to be beneficial as both enzymes have been shown to modulate cholinergic neurotransmission which is key in the pathology of AD [15]. Moreover, in the later stages of AD, BuChE is responsible for the majority of ACh metabolism [15].

Synergism was also identified between four of the flavan-3-ol compounds for AChE inhibition and to a lesser extent, BuChE inhibition. This is of importance because these compounds are likely to be consumed in combination since they are all found in green tea [29]. The varying kinetics of AChE inhibition produced by EGC (uncompetitive) and EGCG (competitive) may account for some of this synergism. Synergy is a known phenomenon in natural product extracts [30].

However, despite some of the novel compounds achieving statistically significant inhibition of cholinesterase enzymes, Gal (the current gold-standard of treatment) was able to produce IC_50_ values much more potent than all of the novel compounds in all experiments conducted.

An IC_50_ value of 0.466 µM was calculated for Gal inhibition of AChE. This is substantially different from a previously calculated value in another study of 4 µM [31]. This discrepancy may be accounted for by differing experimental procedures. Differences in the type of AChE used are likely to change calculated IC_50_ values. Moreover, the same protocol was used for each experiment in this investigation, meaning that the comparisons made between the compounds under investigation are still valid.

Gal was found to competitively inhibit AChE, a finding which is consistent with existing literature [18]. However, in this experiment, inhibition of BuChE by Gal was found to be mixed. This finding may again be a result of different experimental procedures.

### 4.2. Bioavailability and Cellular Accumulation of Flavan-3-Ols

A key aim in the development of new cholinesterase inhibitors is to develop compounds which have side-effect profiles superior to that of Gal. Therefore, assessment of flavan-3-ol compounds in vivo is also important for the identification of any potential side-effects. If flavan-3-ol compounds are shown to have a favourable side-effect profile along with cholinesterase inhibition, they may be an attractive alternative to Gal, with the reduced potency of flavan-3-ols compared to Gal being offset by a more desirable side-effect profile.

Studies have shown that concentrations of flavan-3-ols between 1–100 µmol/L in human cells are sufficient to produce benefits such as antioxidation and anti-inflammation [32]. However, oral administration, the consequent metabolism and limited passive absorption from the gastrointestinal tract, results in flavan-3-ol compounds being able to produce only very low plasma concentrations, typically below the micromolar range [32]. These factors combined with an estimated blood–brain–barrier (BBB) penetrative ability of only 2.8% for EGCG in mice [25] suggests only very small concentrations of flavan-3-ols are likely to be able to reach cholinergic neurons in the brain. However, differences in the BBB permeability to flavan-3-ols may exist between mice and humans and this highlights the need for in vivo human testing.

Despite the significant anti-cholinesterase properties in vitro of EGCG shown, concentrations achieved naturally inside human cells are unlikely to be sufficient to replicate this significant cholinesterase inhibition. Consequently, Gal appears likely to remain the treatment of choice unless novel methods of drug delivery can be developed. Nano delivery has shown some promise and could be used to enhance flavan-3-ol bioavailability [33].

Another alternative could be to develop new formulations for flavan-3-ol delivery. It has been shown that the formulation of flavan-3-ols with ascorbic acid (vitamin C) and sucrose resulted in enhanced bioavailability [34]. Increasing bioavailability is vital in ensuring therapeutic levels of flavan-3-ol compounds can reach the cholinergic neurons for effective cholinesterase inhibition to be achieved. In addition, ascorbic acid has also been shown to counteract the iron depletion and reduced absorption of nutrients from the gastrointestinal tract that can result from the intake of large quantities of dietary flavan-3-ols [35].

An understanding of the genetic polymorphisms affecting the expression and function of the transporters and enzymes which determine the bioavailability of flavan-3-ols is also important. The DTDST transporter has been identified as a relevant transporter in the uptake of flavan-3-ol compounds from the intestines in mice [36]. Functionally relevant polymorphisms in the genes encoding for transporters such as DTDST can lead to altered pharmacokinetic profiles for flavan-3-ol compounds, which in-turn may limit the efficacy of these compounds to specific groups of patients.

One of the limitations of this study was the use of L-B plots as these can skew the results from experiments. This is because taking the reciprocal of results causes the magnification of errors for smaller values whilst larger values are not affected to the same extent [28]. Despite using electric eel AChE, human CNS AChE shares a large amount of homology with electric eel AChE [37]. Therefore, electric eel AChE can be considered a suitable, more cost-effective alternative for use in the Ellman assay. Equine BuChE shares up to 93.4% sequence homology with human BuChE [38] and this makes equine serum BuChE a suitable model for this experiment.

### 4.3. Alternative Approaches

Recently, greater emphasis has been placed on prevention of AD in the first place, especially since the onset of the disease is currently thought to start many years before patients present with symptoms [39]. Reductions in AD incidence have been recorded in the UK over recent years and it is hypothesised that these reductions are attributable to decreased exposure to well-established AD risk factors [40]. Therefore, despite the promise that flavan-3-ols show as novel cholinesterase inhibitors for the treatment of AD, they could be evaluated in the context of preventative medication, possibly being used to mitigate the negative effects of exposure to AD risk factors.

Finally, the combination of this preventative treatment with the use of models which can assign individuals a hazard score according to their genetic profile [41], predicting the likelihood of them developing AD, should allow further reductions in the incidence of AD to be achieved.

## 5. Conclusions

Treatment with natural plant-based compounds such as flavan-3-ols could be used to provide more holistic benefits, targeting age-associated memory impairment or multiple areas of AD pathology. This is in contrast to Gal which produces mainly cholinesterase inhibition with troublesome side-effects [19].

The anti-inflammatory and antioxidative properties of flavan-3-ol compounds [32], combined with significant cholinesterase inhibitory activity demonstrated by EGCG in this study, results in an attractive group of compounds which could be recommended for the treatment of AD symptoms. Further investigation of EGCG and other related flavan-3-ols in vivo is suggested to gain a fuller understanding of the properties of these compounds.

## Figures and Tables

**Figure 1 nutrients-12-01090-f001:**
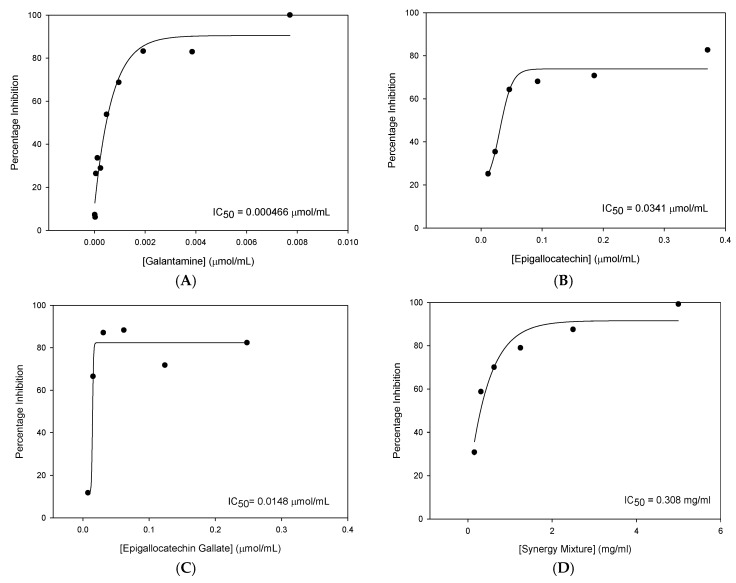
IC_50_ plots of AChE inhibition: concentration against percentage inhibition for each of the active compounds tested. (**A**) galantamine, (**B**) epigallocatechin, (**C**) epigallocatechin-3-gallate and (**D**) synergy mixture.

**Figure 2 nutrients-12-01090-f002:**
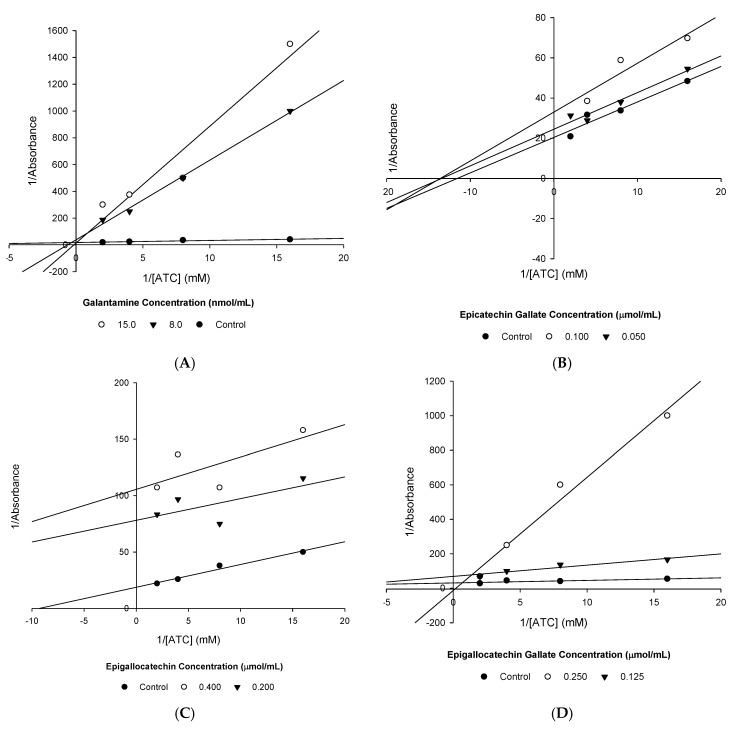
Lineweaver–Burk plots: (**A**) Gal, (**B**) ECG, (**C**) EGC and (**D**) EGCG.

**Figure 3 nutrients-12-01090-f003:**
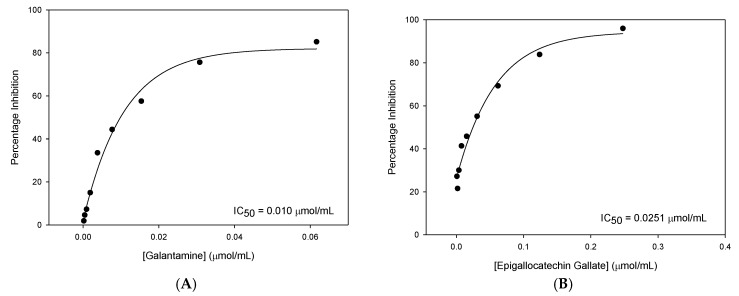
IC_50_ plots for BuChE inhibition: concentration against percentage inhibition for each of the active compounds tested. (**A**) Gal and (**B**) EGCG.

**Figure 4 nutrients-12-01090-f004:**
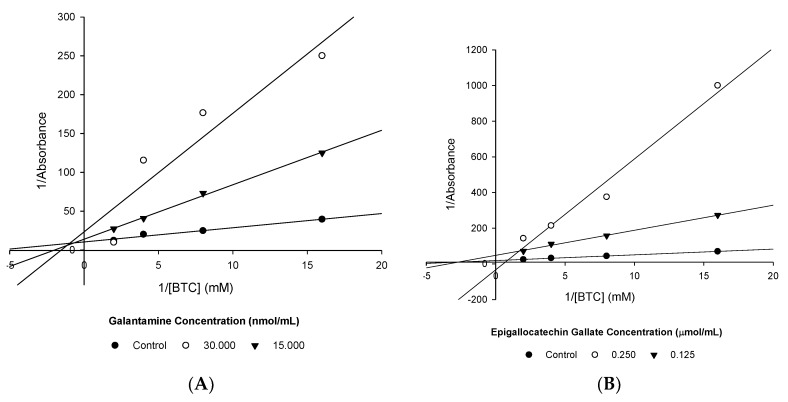
L-B plots of BuChE inhibition. (**A**) Gal and (**B**) EGCG.

**Table 1 nutrients-12-01090-t001:** Compound summary table: an overview of the chemicals assessed in this study with structures and notes for each shown. Chemical structures were produced using ChemDraw^TM^.

Compound		Structure	Notes
	Galantamine	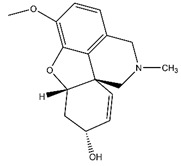	Gold-standard for AChE inhibition [19] Approved for the treatment of mild AD symptoms [19] Poor side-effect profile
**Flavan-3-ols**	Epicatechin Gallate	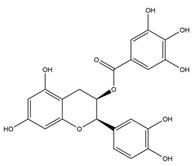	These flavanol compounds are found predominantly in green tea [23] Antioxidative and anti-inflammatory properties identified [21] Dietary intake of catechin polyphenols estimated to be 50 mg/day [24] Blood–brain barrier permeability for these compounds is ~2.8% [25] Consumption of two cups of green tea in 1 h is estimated to produce brain parenchyma concentrations of ~0.01 µM [25] Low concentrations of EGCG in the brain have been reported to slow cognitive decline [23]
	Epicatechin	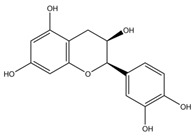	
	Catechin	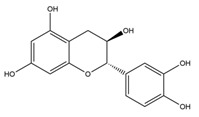	
	Epigallocatechin	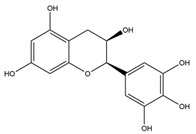	
	Epigallocatechin Gallate	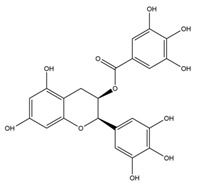	

**Table 2 nutrients-12-01090-t002:** Test stock concentrations: inhibitors were first tested at the highest concentration available to determine if any inhibition was present before further analysis was carried out.

Inhibitor	Stock Concentration * (mg/mL)	Final Assay Concentration (µmol/mL)
Galantamine	0.125	0.008
Epicatechin Gallate	2.000	0.103
Epicatechin	5.000	0.391
Catechin	5.000	0.391
Epigallocatechin	5.000	0.371
Epigallocatechin Gallate	5.000	0.248
Synergy Mixture	1.250	-

* For Flavan-3-ols: Based on solubility limits.

**Table 3 nutrients-12-01090-t003:** AChE inhibition: shows the results for AChE inhibition activity at the highest concentrations available for each compound. Only compounds showing statistically significant levels of inhibition (*p* < 0.01) were tested further.

Compound	Concentration (mg/mL)	Final Assay Concentration (µmol/mL)	AChE Inhibition (%)	*p*-Value
Galantamine	0.125	0.008	93.79	*p* = 0.000
Epicatechin Gallate	2.000	0.103	37.14	*p* = 0.001
Epicatechin	5.000	0.391	13.48	*p* = 0.018
Catechin	5.000	0.391	6.45	*p* = 0.479
Epigallocatechin	5.000	0.371	82.65	*p* = 0.000
Epigallocatechin Gallate	5.000	0.248	82.35	*p* = 0.000
Synergy Mixture	1.250	-	99.22	*p* = 0.000

**Table 4 nutrients-12-01090-t004:** Active compound IC50 value comparisons.

Compound	AChE IC_50_ Value (μmol/mL)	Fold Difference Relative to Galantamine
Galantamine	0.000466	1.0
Epigallocatechin	0.034100	73.2
Epigallocatechin Gallate	0.014800	31.8

**Table 5 nutrients-12-01090-t005:** Synergy mixture IC_50_ comparisons.

Compound	AChE IC_50_ Value (mg/mL)	Fold Difference Relative to Galantamine
Galantamine	0.008	1.0
Synergy Mixture	0.308	40.8

**Table 6 nutrients-12-01090-t006:** Threshold concentrations required for AChE inhibition.

Compound	Threshold [Inhibitor](μmol/mL)
Galantamine	0.00048
Epigallocatechin	0.02300
Epigallocatechin Gallate	0.03100

**Table 7 nutrients-12-01090-t007:** Kinetic coefficients.

Inhibitor	[Inhibitor] (mg/mL)	[Inhibitor] (µmol/mL)	V_MAX_ (mM)	K_M_ (mM)
Galantamine	0.000	0.000	0.055	0.083
	0.125	0.008	0.026	1.564
	0.250	0.015	0.071	6.177
Epicatechin Gallate	0.000	0.000	0.049	0.087
	1.000	0.051	0.041	0.075
	2.000	0.103	0.030	0.074
Epigallocatechin	0.000	0.000	0.053	0.107
	2.500	0.186	0.013	0.025
	5.000	0.371	0.009	0.027
Epigallocatechin Gallate	0.000	0.000	0.032	0.048
	2.500	0.124	0.014	0.094
	5.000	0.248	0.081	5.305

**Table 8 nutrients-12-01090-t008:** BuChE inhibition: initial tests for BuChE inhibition at the highest concentrations available for each compound. Only compounds showing statistically significant levels of inhibition (p < 0.01) were tested further.

Compound	Concentration (mg/mL)	Final Assay Concentration (µmol/mL)	BuChE Inhibition (%)	*p*-Value
Galantamine	0.125	0.008	93.79	*p* = 0.000
Epicatechin Gallate	2.000	0.103	45.65	*p* = 0.001
Epicatechin	5.000	0.391	11.62	*p* = 0.272
Catechin	5.000	0.391	18.26	*p* = 0.097
Epigallocatechin	5.000	0.371	47.72	*p* = 0.001
Epigallocatechin Gallate	5.000	0.248	89.64	*p* = 0.000
Synergy Mixture	1.250	-	56.02	*p* = 0.000

**Table 9 nutrients-12-01090-t009:** BuChE inhibition.

Compound	BuChE IC_50_ Value (μmol/mL)	Fold Difference Relative to Galantamine
Galantamine	0.01000	1.00
Epigallocatechin Gallate	0.02510	2.51

**Table 10 nutrients-12-01090-t010:** Threshold concentrations required for BuChE inhibition.

Compound	Threshold [Inhibitor] (μmol/mL)
Galantamine	0.0313
Epigallocatechin Gallate	<0.001

**Table 11 nutrients-12-01090-t011:** Kinetic coefficients.

[Inhibitor] (mg/mL)	[Inhibitor] (mg/mL)	[Inhibitor] (µmol/mL)	V_MAX_ (mM)	K_M_ (mM)
Galantamine	0.000	0.000	0.096	0.175
	0.125	0.008	0.072	0.501
	0.500	0.031	0.043	0.651
Epigallocatechin Gallate	0.000	0.000	0.054	0.172
	0.125	0.124	0.021	0.293
	0.250	0.248	0.030	1.863

**Table 12 nutrients-12-01090-t012:** IC_50_ comparison for AChE and BuChE.

Compound	AChE IC_50_ Value (μmol/mL)	BuChE IC_50_ Value (μmol/mL)	Selectivity for AChE
Galantamine	0.000466	0.01000	21.5
Epigallocatechin Gallate	0.014800	0.02510	1.7

**Table 13 nutrients-12-01090-t013:** Synergy mixture cholinesterase inhibition comparison: the synergy mixture was much more effective at inhibiting AChE although relevant inhibition of BuChE was still detected.

Compound	AChE Inhibition (%)	BuChE Inhibition (%)
Synergy Mixture	99.22	56.02

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
