# Peer review of "Comparative Kinetics of Acetyl- and Butyryl-Cholinesterase Inhibition by Green Tea Catechins|Relevance to the Symptomatic Treatment of Alzheimer’s Disease"

_nutrients, 2020, doi:10.3390/nu12041090_

Round 1

Reviewer 1 Report

The manuscript submitted by Okello and Mather explains the utility of green tea catechizes against AD. The study is planned and presented well enough. I would suggest acceptance after minor improvement in the article;

  1. Materials should be descriptive including catalog numbers/source of reagents used in the study.
  2. introduction section is too lengthy and must be reduced.
  3. the study is submitted in chapter/thesis like format. the headings and subheadings in introduction/discussions should be removed. Text must be streamlined for uninterrupted reading.

Reviewer 2 Report

Reviewer comment:

- The manuscript entitled ,,Comparative Kinetics of Acetyl- and Butyryl-cholinesterase Inhibition by Green Tea Catechins | Relevance to the Symptomatic Treatment of Alzheimer’s disease'' provides interesting data on this topic.

This is a nicely written contribution based on clearly described experiments. The topic of this study is interesting and in my opinion it could be interesting for a reasonable number of scientists.

Further in vitro, in vivo and clinical studies are needed to better investigate the processes described and to learn about side effects.

I have a few minor editorial comments:

- examples in line:

66  significant importance (10). bad distance;   significant importance(10). good distance;

72  prevalent (11).

80 with AD (13).

- please also check the remaining spacing for citations in the remaining text ….

- please also check the remaining intervals in 2.1 Ellman Essay for please also check the remaining intervals in 2.1 Ellman Essay for the descriptions of quantity, concentration, pH, rpm in the text:

5μl or 5 μl, 0.03U/ml or 0.03 U/ml, pH7 or pH 7, pH8 or pH 8, 600rpm or 600 rpm, 30°C or 30 °C.

- lines 141-149 to cross out - error - fragment of instructions for writing the introduction:

141

The introduction should briefly place the study in a broad context and highlight why it is 142 important. It should define the purpose of the work and its significance. The current state of the 143 research field should be reviewed carefully and key publications cited. Please highlight 144 controversial and diverging hypotheses when necessary. Finally, briefly mention the main aim of the 145 work and highlight the principal conclusions. As far as possible, please keep the introduction 146 comprehensible to scientists outside your particular field of research. References should be 147 numbered in order of appearance and indicated by a numeral or numerals in square brackets, e.g., 148 [1] or [2,3], or [4–6]. See the end of the document for further details on references.

- line 225: inFigure 1. or in Figure 1.

- line 265: inFigure 3. or in Figure 3.

- Figure 1D: IC50 = 0.308mg/ml or IC50 = 0.308 mg/ml

- line 307:  0.466μM or 0.466 μM

- line 308:  4μM or 4 μM

- page correction - References:

- line 402:

6 Corder EH, Saunders AM, Strittmatter WJ, Schmechel DE, Gaskell PC, Small GW, Roses AD,  Haines JL, Pericak-Vance MA. Gene dose of apolipoprotein E type 4 allele and the risk of  Alzheimer's disease in late onset families. Science. 1993;261:921-923.

- line 430:

14 Barbier P, Zejneli O, Martinho M, Lasorsa A, Belle V, Smet-Nocca C, Tsvetkov PO, Devred F,  Landrieu I. Role of Tau as a Microtubule-Associated Protein: Structural and Functional  Aspects. Frontiers in Aging Neuroscience. 2019;11:204:1-14.

- line 508:

36 Cai Z-Y, Li X-M, Liang J-P, Xiang L-P, Wang K-R, Shi Y-L, Yang R, Shi M, Ye J-H, Lu J-L, et al.  Bioavailability of Tea Catechins and Its Improvement. Molecules. 2018;23:2346:1-18.

- line 524

40 Ishii S, Kitazawa H, Mori T, Kirino A, Nakamura S, Osaki N, Shimotoyodome A, Tamai I.  Identification of the Catechin Uptake Transporter Responsible for Intestinal Absorption of  Epigallocatechin Gallate in Mice. Scientific Reports. 2019;9:11014:1-10.

- line 534

43 Crous-Bou M, Minguillón C, Gramunt N, Molinuevo JL. Alzheimer's disease prevention:  from risk factors to early intervention. Alzheimers Res Ther. 2017;9:71:1-9.

After a slight correction, I suggest acceptance of this paper.

Reviewer 3 Report

This is an interesting writing on the effects of flavan-3-ols found in green tea on BuChE and AChE activities and speculation on their potentialities in neurodegenerative diseases such as Alzheimer disease.

The results are worth testing in vivo and in vitro in appropriate models.

Theare are few suggestions to be considered:

  1. it is unclear the selection of inibitors in the so called "synergy mixture" and it is not explained why the authors did not consider to combine EGC and EGCG rather than four
  2. It should be explained why they selected the given concentrations in table 3
  3. The authors should consider  a final paragraph on the potentialities in treating patients with natural supplements in normal and diseased aging

Minor:

Remove page 5, ln142-149 8guess is a typo)
